# Modelling of performance prediction by analysis of elite swimmers' anthropometry, peak performance age and age-related performance progression

Amir Nazari Mehrabi[1], Hamoon Imani[1], Omid Khademnoe[2], Mina Khantan[3], Tommy R. Lundberg[4,5], Ali Gorzi[1,4]*

1 Department of Sport Sciences, University of Zanjan, Zanjan, Iran, 2 Department of Statistics, University of Zanjan, Zanjan, Iran, 3 Faculty of Sport Sciences and Health, Shahid Beheshti University, Tehran, Iran, 4 Division of Clinical Physiology, Department of Laboratory Medicine, Karolinska Institutet, Stockholm, Sweden, 5 Unit of Clinical Physiology, Karolinska University Hospital, Stockholm, Sweden

* ali.gorzi.2@ki.se

## Abstract

In this study, anthropometry, age-related performance progression and peak performance age (PPA) in elite swimmers were analysed to develop a model to predict peak performance. The best seasonal performances of the world's all-time top 20 male and female swimmers in 5 strokes/styles (FS: freestyle; BK: backstroke; BT: breaststroke; BF: butterfly; and MD: medley) and 17 individual events were considered. An event- and sex-specific model using dynamic panel data methods was used to calculate and present 95% confidence bands to formulate performance trends. We also analysed the historical changes in PPA, height and body mass by dividing these 20 top swimmers into two groups based on their YOB: former (n = 10) and recent (n = 10) swimmers. The height of male FS swimmers was significantly greater than that of BF and MD swimmers, and BK swimmers was greater than that of MD swimmers, and among females, height of MD swimmers was significantly smaller than that of FS, BK and BF swimmers. The PPA of BT swimmers was significantly higher than that of FS, BK and BF swimmers in males, and BF swimmers was significantly higher than that of MD swimmers in females. Both male and female more recent swimmers were shorter, lighter and, in particular, younger than their former counterparts in most events. Performance over the preceding 1 year in all events and 2 years in men's 50m BT and women's 100m BK, 100m BT, 200m BT and 200m BF, and weight in 100m BK were important for predicting future performance. Our models provide coaches with a practical tool for predicting PPA, performance records and appropriate benchmarks at different ages, which can be useful for talent identification, goal setting and evaluation of performance progression.

**Data availability statement:** All relevant data are available at the following links: - https://www.worldaquatics.com/athletes?gender=&-discipline=&nationality=&name= - https://www.swimrankings.net/index.php?page=rankingDe-tail&clubId=1&gender=1&course=LCM&age-group=0&stroke=0&season=-1.

**Funding:** The author(s) received no specific funding for this work.

**Competing interests:** The authors have declared that no competing interests exist.

## Introduction

Swimming performance is influenced by several factors, including technique, strength, power and body composition. Previous research has shown the importance of anthropometric characteristics such as body size, limb length and muscle mass in achieving greater propulsion and speed, which contribute to success in swimming at the elite level [1]. Understanding the factors that contribute to success in swimming, including age-related performance development, peak performance age (PPA) and anthropometric characteristics, is critical for coaches, athletes and sport scientists involved in talent identification and development and can also help set realistic expectations for young athletes.

Despite the vast resources put into talent identification systems worldwide, most attempts to predict the future potential of athletes fail [2]. There is generally little correspondence between performance in youth and senior age [3], and peak per-formers in adulthood are largely a different population from peak performers in youth [4]. This is partly because factors such as relative age or relative maturity effects can cloud performance assessments of young athletes and confuse advanced physique with future potential [5], but also because elite athletes often have multi-sport back-grounds and later participate in the most structured talent development programs [6]. Therefore, by establishing acceptable performance benchmarks at different age levels, and considering the sport-specific requirements [7–9], coaches and young athletes can create more informed long-term development plans and set appropriate goals for an athlete's entire career [10].

Previous studies in athletics have provided valuable insights into performance trends and PPA in various events. A discipline- and sex-specific model was used to analyse performance trends by employing dynamic panel data methods [11]. For example, it was reported that performance in the previous 1–2 years, height, and weight could be used to predict future records, but with different coefficients and different factors included in the model depending on the discipline. However, the specific requirements of success in swimming, such as the number of races swum per year throughout the athletic career [12], muscle quality [9], and specific tests related to performance prediction [13], justify a specific study of these aspects in the context of aquatic sports. In addition, analysing performance development and PPA separately for male and female swimmers can reveal important differences in development patterns between the sexes [7,14], which is crucial for adapting train-ing and competition strategies. For example, the correlation between anthropomet-ric factors and performance may be lower in young girls than in boys [15] or even absent [7], which is likely due to the generally less favorable physiological changes in girls compared to boys during puberty [16]. In addition, despite the mostly similar duration of the peak performance window in men and women, the PPA may differ across sports [17,18].

A previous study attempted to model the performance development of sub-elite adolescent swimmers over a variety of individual distances [19]. The authors iden-tified different rates of improvement in different events, which could be useful for

monitoring and assessing benchmarks; however, the study did not follow athletes through to their peak performances in adulthood. It is therefore unknown to what extent performance at younger ages reflect performance level in adulthood. Instead, by establishing confidence limits for acceptable performance at different stages of development in athletes who are known to have reached elite-level performance, coaches and talent developers can be provided with a tool to guide the long-term development of athletes. In addition, analysing the height and weight of elite athletes over time can reveal potential shifts in the anthropometric profile of elite swimmers that may have implications for talent identification, prediction models and training methods [20].

There is a reciprocal relationship between the development of skeletal muscle, bone, nervous, cardiovascular, hormonal, cardiorespiratory and other systems and organs with exercise training [21]. On the other hand, aging, as an inevitable process, poses a challenge to the physiological capacities and performance of athletes. Swimming is well known as a sport in which PPA occurs relatively early [22], but the disadvantages of early specialization [11] along with the recently observed high quality performance of older swimmers, present an interesting conflict. Therefore, how age affects peak performance, when performance declines in different swimming events occur, and how the characteristics of former and more recent top swimmers have changed over time, possibly driven by secular trends in pubertal timing [23–25], is an interesting area of research and could provide important information for the development of performance prediction models in swimming [11,26–28].

In the present study, we aimed to analyse age-related performance progression and PPA in the all-time top swimmers in the world and to use a model to predict peak performance. By establishing confidence limits for acceptable performance at different developmental stages, this study could be a useful tool for coaches to predict peak performance, PPA and desirable performance levels during development. We also examined the trends in the anthropometric parameters of these world-class swimmers over time.

## Methods

### Data sample

The data included the all-time top 20 list of performances in swimming competitions, which included about 680 athletes (340 M: 340 W). Their season-best performances from start (junior age) to retirement were collected from FINA (recently changed to world aquatics) [29] and Swimrankings [30]. Considering the inclusion of some athletes in the top 20 list of two or three events (e.g., 50m, 100m and 200m), less than 20 unique athletes were used for analysis in some events. Chronological age was calculated as year of competition minus year of birth (YOB) and the upper and lower limits of the age (month of YOB and month of competition) were not considered. We also divided these 20 top swimmers into two groups based on their YOB: former (n = 10) and recent (n = 10) swimmers (e.g., in the 50 m freestyle event for men, the 10 swimmers with a YOB between 1979 and 1987 were classified as former, while the remaining 10 swimmers with a YOB between 1988 and 1999 were classified as recent), and analysed the historical changes in PPA, height and weight.

All data were categorized into five main groups of strokes/styles and the 17 official Olympic events for men and women (1- freestyle; **FS**: 50, 100, 200, 400, 800, 1500m; 2- backstroke; **BK**: 50, 100, 200m; 3- breaststroke; **BT**: 50, 100, 200m; 4- butterfly; **BF**: 50, 100, 200m; and 5- medley; **MD**: 200 and 400m). The relay events were not included in this study. Standard 50m indoor pool records were included to avoid the effects of this factor on the results. All official rules and regulations were considered, and the available data were followed until January 2024 or until the retirement age of the swimmers. Records are limited to the individual best performance (season best) for each event in the years listed.

For modelling, we used $R_t$ as the predicted record at a given age, $R_{t-1}$ as the best personal record one year ago, $R_{t-2}$ as the best personal record two years ago, $m.R_t$ as the average (mean) of the records of the all-time top lists at given age (Table 1), H as the height of the athlete at adulthood and W, as the weight of the athlete at adulthood. All records considered in the second and centesimal and outlier data from each age group were excluded by boxplot (e.g., in the men's 200 m freestyle, the records ranged from 104.06 to 109.23 seconds, and one outlier with a time of 111.26 seconds was

**Table 1. Performance progression (M±SD) of all-time top 20 swimmers in all 17 events for Men and Women.**

| Age/disc | 50 FS M (s) | 50 FS W (s) | 100 FS M (s) | 100 FS W (s) | 200 FS M (s) | 200 FS W (s) | 400 FS M (s) | 400 FS W (s) | 800 FS M (s) | 800 FS W (s) |
|---|---|---|---|---|---|---|---|---|---|---|
| 10 | 30.54 | | | 71.33 | | 151.21±7.18 | | 306.23 | | |
| 11 | | | 70.36±2.63 | 69.79±0.45 | 144.79 | 134.62 | 293.57 | 284.14 | | |
| 12 | | 26.84±1.33 | 65.04±0.23 | 62.75±2.03 | | 134.96±5.77 | 273.71±3.03 | 268.7±8.17 | 557.74 | |
| 13 | 28.46±1.65 | 27.45±0.65 | 61.03±0.31 | 60.27±1.65 | 132.09±2.48 | 129.23±4.12 | 272.57±5.68 | 268.14±15.7 | 544.70±15.19 | 538.72±23.14 |
| 14 | 27.16±3.32 | 26.32±0.81 | 57.64±2.69 | 57.48±1.68 | 123.72±1.34 | 123±3.28 | 250.07±3.23 | 258.92±7.86 | 530.81±9.17 | 533.17±14.64 |
| 15 | 24.85±1.51 | 25.99±0.70 | 52.32±2.15 | 56.20±1.57 | 115.34±3.37 | 121.31±3.50 | 239.57±7.18 | 251.90±5.87 | 514.07±14.56 | 521.51±17.65 |
| 16 | 24.38±1.43 | 25.59±0.62 | 51.31±1.38 | 55.63±2.04 | 111.50±3.29 | 118.27±1.93 | 233.55±4.90 | 251.17±7.37 | 498.22±13.06 | 513.05±10.55 |
| 17 | 23.24±0.81 | 25.19±0.60 | 49.31±1.15 | 55.10±1.76 | 110.00±3.08 | 118.01±2.43 | 233.62±6.69 | 249.20±5.81 | 487.63±9.47 | 510.16±10.80 |
| 18 | 22.91±0.64 | 25.23±0.75 | 49.37±1.65 | 54.48±1.40 | 107.73±2.14 | 117.06±2.64 | 229.56±4.31 | 246.65±5.24 | 481.27±8.97 | 506.6±10.97 |
| 19 | 22.57±0.59 | 24.86±0.47 | 49.44±0.99 | 54.47±1.18 | 106.90±1.51 | 116.59±1.91 | 227.51±3.94 | 244.84±3.89 | 474.26±10.07 | 504.80±9.44 |
| 20 | 22.41±0.66 | 24.75±0.56 | 48.96±1.05 | 53.88±1.21 | 106.25±1.35 | 116.63±1.83 | 226.55±3.88 | 244.82±3.29 | 473.61±9.75 | 505.16±8.24 |
| 21 | 22.05±0.57 | 24.69±0.65 | 48.58±1.23 | 53.62±0.88 | 106.02±1.30 | 115.95±1.81 | 225.02±3.27 | 244.14±3.91 | 467.63±6.32 | 502.26±8.33 |
| 22 | 21.86±0.51 | 24.53±0.57 | 48.26±1.01 | 53.31±0.65 | 105.42±1.07 | 116.10±1.88 | 225.33±3.90 | 242.79±2.80 | 465.47±5.39 | 500.16±5.26 |
| 23 | 21.75±0.44 | 24.59±0.71 | 48.08±0.85 | 53.14±0.57 | 105.57±1.67 | 115.75±1.57 | 224.69±2.34 | 244.45±5.43 | 465.72±5.17 | 503.06±8.90 |
| 24 | 21.91±0.61 | 24.28±0.44 | 48.26±0.70 | 53.09±0.89 | 105.43±1.07 | 115.90±1.19 | 225.90±3.15 | 243.98±3.98 | 467.77±6.39 | 502.44±8.46 |
| 25 | 21.67±0.36 | 24.38±0.44 | 48.19±0.67 | 53.34±0.60 | 105.83±0.82 | 115.73±0.87 | 225.16±3.90 | 246.65±3.96 | 466.32±7.75 | 502.20±7.82 |
| 26 | 21.82±0.52 | 24.62±0.74 | 48.16±0.76 | 53.19±1.28 | 105.38±0.87 | 117.40±2.83 | 226.09±3.46 | 245.06±1.90 | 468.40±8.42 | 505.02±8.48 |
| 27 | 21.89±0.36 | 24.36±0.39 | 48.42±0.39 | 52.97±0.61 | 106.39±1.52 | 116.03±1.58 | 226.40±4.11 | 247.24±2.83 | 467.24±2.80 | 500.33±5.24 |
| 28 | 21.81±0.56 | 24.44±0.34 | 48.06±0.96 | 53.56±1.18 | 105.54±0.55 | 116.59±3.21 | 225.49±2.60 | 250.06±2.68 | 466.98±3.96 | 507.84±8.21 |
| 29 | 21.81±0.30 | 24.36±0.37 | 48.54±0.56 | 53.17±0.55 | 107.16±2.29 | 115.68±0.88 | 229.06±3.81 | 250.11±1.84 | 474.40±5.14 | 508.67±5.19 |
| 30 | 21.66±0.26 | 24.36±0.28 | 49.16±0.63 | 53.51±0.21 | 107.66±1.82 | 116.14±0.19 | 228.75±3.10 | 250.96±2.20 | | 511.06 |
| 31 | 21.86±0.22 | 24.33±0.35 | 48.97±0.60 | 53.18±0.07 | 106.78±2.01 | 115.55±1.16 | 228.81±2.48 | 253.24±3.61 | 476 | |
| 32 | 21.86±0.30 | 24.53±0.57 | 48.62 | 53.05 | 106.62 | 116.96±0.86 | 232.86±0.37 | 263.12 | 483.21 | |
| 33 | 21.89±0.38 | 24.40±0.19 | 50.16±2.53 | 53.45 | | 116.79±1.24 | 233.49 | 263.65 | | |
| 34 | 22.35±0.32 | 24.14 | 50.24 | 52.79 | | 119.05 | | | | |
| 35 | 22.54±0.96 | 24.5 | 50.79 | | 110.25 | | 233.39 | | | |
| 36 | 22.61±0.49 | | | | 112.05 | | 227.75 | | | |
| 37 | | 24.99 | 49.46 | | 109.23 | | | | | |
| 38 | 22.68 | 25.29 | 47.99 | | | | | | | |
| 39 | 22.90 | 25.69±1.37 | | | | | | | | |
| 40 | 22.64 | 24.53 | | | | | | | | |
| 41 | | 24.07 | | | | | | | | |
| 42 | | 24.43 | | | | | | | | |

| Age/disc | 1500 FS M (s) | 1500 FS W (s) | 50 BK M (s) | 50 BK W (s) | 100 BK M (s) | 100 BK W (s) | 200 BK M (s) | 200 BK W (s) | 50 BT M (s) | 50 BT W (s) |
|---|---|---|---|---|---|---|---|---|---|---|
| 10 | | | 36 | | | 85.85±9.99 | | | 42.37±3.76 | 42.8 |
| 11 | | | | 32.82 | 72.07 | 72.88±0.90 | | 155.53±1.96 | | 39.25±4.39 |
| 12 | | | 34.43 | 37.86±2.28 | 72.74 | 71.66±6.18 | | 149.61±13.01 | | 36.14±2.59 |
| 13 | 1086.45 | 992.88 | 32.06±0.64 | 34.04±1.30 | 66.03±0.75 | 66.77±3.14 | 135.67±2.74 | 140.70±5.25 | 34.59±1.49 | 35.13±3.85 |
| 14 | 1063.44±39.85 | 1019.05±30.90 | 29.27±1.68 | 31.29±1.49 | 60.67±3.08 | 64.02±3.00 | 127.89±4.31 | 136±8.23 | 32.22±1.01 | 32.96±1.95 |
| 15 | 964.71±22.75 | 1005.99±25.11 | 27.10±0.81 | 29.65±1.16 | 58.79±2.54 | 61.73±2.11 | 127.95±5.43 | 133.31±5.29 | 31.02±2.01 | 32.37±1.88 |
| 16 | 949.04±22.32 | 985.78±30.32 | 25.92±0.74 | 29.20±0.96 | 56.35±1.84 | 61.15±1.73 | 121.82±2.73 | 131.60±5.43 | 30.17±1.32 | 31.56±1.52 |
| 17 | 921.20±18.60 | 979.09±28.68 | 25.73±0.60 | 28.85±0.84 | 55.64±1.66 | 60.61±1.92 | 119.26±2.37 | 131.50±6.45 | 29.24±0.95 | 31.66±1.37 |
| 18 | 911.55±20.18 | 963.63±21.99 | 25.44±0.90 | 28.72±0.76 | 54.87±1.52 | 60.21±1.98 | 117.88±2.06 | 129.91±4.33 | 28.58±0.93 | 31.26±1.09 |
| 19 | 904.08±21.60 | 960.87±14.30 | 25.36±0.73 | 28.18±0.78 | 54.44±1.61 | 59.90±1.69 | 116.57±1.79 | 128.60±2.24 | 28.25±0.94 | 30.75±0.90 |
| 20 | 896.66±17.14 | 954.92±11.74 | 25.15±0.60 | 28.15±0.61 | 54.15±1.41 | 59.88±1.80 | 116.72±2.32 | 127.69±2.21 | 27.93±0.76 | 30.67±0.78 |
| 21 | 886.02±10.27 | 952.91±20.47 | 24.78±0.63 | 28.04±0.75 | 53.58±1.41 | 59.42±1.04 | 116.15±2.02 | 127.91±1.78 | 27.62±0.73 | 30.50±0.83 |
| 22 | 886.69±8.28 | 953.45±9.77 | 24.60±0.40 | 28.06±0.64 | 53.30±1.17 | 59.28±0.80 | 115.73±2.30 | 128±1.68 | 27.28±0.55 | 30.82±0.78 |
| 23 | 885.84±6.44 | 959.11±19.91 | 24.70±0.34 | 27.88±0.42 | 53.29±1.03 | 59.41±1.07 | 114.99±1.85 | 128.76±2.91 | 27.21±0.55 | 30.54±0.45 |

*(Continued)*

**Table 1.** (Continued)

| | | | | | | | | | | |
|---|---|---|---|---|---|---|---|---|---|---|
| 24 | 887.13±8.67 | 960.37±15.38 | 24.69±0.47 | 27.75±0.44 | 53.22±0.73 | 59.44±0.79 | 115.49±2.13 | 127.49±1.41 | 27.19±0.52 | 30.75±0.49 |
| 25 | 890.04±11.78 | 960.49±21.65 | 24.61±0.42 | 27.72±0.34 | 52.71±0.48 | 59.2±0.96 | 115.51±1.55 | 128.36±3.34 | 27.16±0.55 | 30.26±0.58 |
| 26 | 892.62±15.57 | 961.55±9.54 | 01.51±0.26 | 27.65±0.36 | 52.99±0.79 | 58.91±0.40 | 115.80±1.95 | 126.98±1.61 | 27.30±0.43 | 30.14±0.46 |
| 27 | 894.17±8.45 | 948.79±4.15 | 24.71±0.31 | 27.68±0.39 | 53.21±0.86 | 58.97±0.38 | 116.95±2.87 | 128.98±3.70 | 27.05±0.51 | 30.47±0.53 |
| 28 | 883.32±8.87 | 958.90±26.53 | 24.72±0.34 | 27.51±0.33 | 53.12±0.39 | 58.85±0.35 | 115.75±1.49 | 127.98±1.85 | 27.06±0.33 | 30.67±0.41 |
| 29 | 907.89 | 970.38±11.70 | 24.72±0.31 | 27.69±0.26 | 53.81±1.40 | 59.15±0.56 | 116.95±2.91 | 127.15±1.39 | 26.86±0.34 | 30.68±0.72 |
| 30 | | 971.45±9.09 | 24.62±0.28 | 27.61±0.64 | 53.56±1.10 | 59.65±0.24 | 115.87±0.63 | 128.52±2.24 | 27.04±0.38 | 30.53 |
| 31 | | 969.7 | 25.12±0.43 | 27.93±0.56 | 53.77±2.36 | 60.37±1.22 | 117.22±1.28 | 129.95 | 26.98±0.65 | 30.31 |
| 32 | 907.78 | | 24.82±0.73 | 27.23 | 54.02±2.12 | 61.88 | 118.22±2.44 | 130.95±1.32 | 27.03±0.27 | 30.44 |
| 33 | | | 24.97 | | 53.44±0.77 | | 117.13 | 128.8 | 26.83±0.41 | 31.69 |
| 34 | | | | | 54.73±2.70 | | 120.55±0.30 | | 26.76±0.38 | |
| 35 | | | | | 53.54 | | 119.26 | | 27.23±0.27 | |
| 36 | | | | | 53.18 | | 123.83 | | 26.83±0.12 | |

| Age/disc | 100 BT M (s) | 100 BT W (s) | 200 BT M (s) | 200 BT W (s) | 50 BF M (s) | 50 BF W (s) | 100 BF M (s) | 100 BF W (s) | 200 BF M (s) | 200 BF W (s) |
|---|---|---|---|---|---|---|---|---|---|---|---|
| 10 | | 92.9±1.48 | | | 35.80±2.50 | | 78.84±2.08 | 87.03±13.9 | | |
| 11 | | 83.6 | | | 34.35±2.03 | | 72.67±5.90 | 76.75±5.53 | | 150.21 |
| 12 | 78.11 | 81.18±4.13 | 155.10 | 177.55±24.23 | 32.37±1.32 | 31.59±1.48 | 68.10±2.84 | 71.82±4.57 | 156.07±18.67 | 140.51 |
| 13 | 79.43±11.66 | 72.36±2.34 | 141.39±4.86 | 164.98±9.27 | 30.81±2.97 | 30.68±1.81 | 66.31±4.79 | 64.52±2.13 | 144.00±7.11 | 139.22±6.62 |
| 14 | 68.04±5.13 | 70.78±1.89 | 144.83±11.95 | 155.33±4.76 | 26.74±2.15 | 28.76±1.59 | 59.74±3.88 | 61.20±1.47 | 136.41±7.60 | 136.65±4.95 |
| 15 | 66.87±4.31 | 69.81±2.95 | 143.07±10.07 | 153.02±7.32 | 26.62±1.99 | 27.54±1.15 | 56.13±3.00 | 60.36±2.17 | 124.98±7.31 | 134.89±4.24 |
| 16 | 64.45±2.62 | 70.27±3.14 | 138.72±8.52 | 151.91±6.76 | 24.64±0.66 | 26.83±0.72 | 54.47±2.39 | 58.98±2.19 | 120.50±5.56 | 131.80±4.31 |
| 17 | 63.73±2.30 | 68.46±2.36 | 135.19±4.47 | 150.37±6.17 | 24.45±0.93 | 26.45±0.69 | 54.02±1.80 | 58.79±1.60 | 117.82±3.14 | 131.08±4.66 |
| 18 | 62.69±1.86 | 67.92±1.84 | 133.04±4.76 | 147.34±5.06 | 24.31±0.87 | 26.32±0.75 | 53.39±1.79 | 58.83±1.67 | 116.50±2.47 | 128.55±3.06 |
| 19 | 62.29±2.38 | 67.89±1.97 | 131.63±4.60 | 147.81±5.43 | 23.82±0.68 | 26.50±0.67 | 52.41±1.46 | 58.11±1.52 | 116.21±2.50 | 128.50±3.56 |
| 20 | 61.07±1.61 | 67.52±1.98 | 130.32±3.45 | 145.57±3.94 | 23.82±0.66 | 26.25±0.66 | 52.23±1.18 | 57.77±1.46 | 115.54±2.24 | 127.33±2.70 |
| 21 | 60.53±1.35 | 66.85±1.62 | 130.22±2.85 | 143.94±3.69 | 23.51±0.56 | 25.74±0.51 | 51.44±1.15 | 57.90±1.18 | 115.13±2.07 | 126.84±2.17 |
| 22 | 59.71±1.09 | 66.67±1.67 | 129.78±2.90 | 143.21±2.46 | 23.51±0.45 | 25.84±0.54 | 51.24±1.01 | 57.53±1.62 | 115.46±2.55 | 127.10±2.25 |
| 23 | 59.60±1.06 | 66.37±1.44 | 129.20±2.22 | 144.06±2.80 | 23.16±0.41 | 26.06±0.87 | 51.13±0.83 | 57.39±1.65 | 114.95±1.83 | 127.46±2.31 |
| 24 | 59.35±0.90 | 66.53±1.41 | 128.84±1.68 | 143.22±2.66 | 23.17±0.41 | 25.72±0.67 | 51.68±0.86 | 57.09±1.03 | 114.55±1.51 | 126.95±1.56 |
| 25 | 59.15±0.86 | 66.36±1.20 | 128.95±2.02 | 142.76±2.58 | 23.17±0.41 | 25.86±0.52 | 51.38±1.19 | 57.03±1.03 | 115.10±1.50 | 127.31±1.83 |
| 26 | 59.33±0.95 | 65.74±0.64 | 128.32±1.32 | 143.30±2.56 | 23.19±0.49 | 25.68±0.60 | 51.50±0.99 | 57.75±1.41 | 115.36±2.21 | 127.10±1.90 |
| 27 | 59.29±0.94 | 66.20±0.96 | 129.88±1.51 | 142.92±2.48 | 23.38±0.51 | 25.95±0.70 | 51.64±0.76 | 56.79±0.88 | 115.32±1.93 | 126.12±1.11 |
| 28 | 59.23±0.84 | 67.04±0.84 | 129.35±1.46 | 143.15±2.40 | 23.24±0.54 | 25.87±0.56 | 52.21±0.87 | 57.16±0.84 | 115.13±1.27 | 126.81±1.55 |
| 29 | 59.70±0.93 | 67.17±1.43 | 128.70±0.69 | 145.61±3.06 | 23.13±0.26 | 25.58±0.50 | 51.41±0.77 | 57.42±1.03 | 116.13±0.98 | 128.67±1.40 |
| 30 | 60.61±0.95 | | 129.08±1.03 | 147.22 | 23.35±0.35 | 25.45±0.16 | 51.76±1.14 | | 115.88±2.30 | 128.98±2.59 |
| 31 | 62.27±1.30 | | 129.34 | 146.04 | 23.34±0.10 | 25.75±0.69 | 51.47±0.67 | 58.44±1.32 | 116.30±2.91 | 129.39±2.69 |
| 32 | 65.75 | | 132.31 | | 23.08±0.21 | 25.61±0.48 | 51.79 | 59.63±1.50 | 115.75±1.84 | 130.56±2.48 |
| 33 | 66.20 | | 131.05 | | 22.91±0.14 | 25.72±0.21 | 51.65 | 58.74 | 117.38±2.07 | 129.38 |
| 34 | | | | | 23.49±0.34 | 25.73±0.33 | | 58.9 | 116.80±1.40 | |
| 35 | | | | | 23.72±0.66 | 25.58 | | | 119.36 | |
| 36 | | | | | | | | | 118.28 | |
| 37 | | | | | 22.79 | 26.06 | | | | |
| 38 | | | | | | 26.11 | | | | |
| 39 | | | | | 23.12±0.74 | 25.78 | | | | |
| 40 | | | | | 22.95 | | | | | |
| 41 | | | | | | | | | | |
| 42 | | | | | 22.78 | | | | | |

*(Continued)*

**Table 1.** (Continued)

| Age/disc | 200 MD M (s) | 200 MD W (s) | 400 MD M (s) | 400 MD W (s) |
|---|---|---|---|---|
| 10 | 182.77±7.82 | 162.95 | | |
| 11 | 170.04±2.99 | 158.28±6.85 | | 317.51 |
| 12 | 161.96±4.15 | 145.52 | 334.83 | 310.13±7.50 |
| 13 | 150.00±9.53 | 144.16±5.12 | 325.65±16.01 | 294.31±7.94 |
| 14 | 137.77±12.62 | 140.48±5.40 | 287.88±18.55 | 294.75±13.22 |
| 15 | 129.93±7.16 | 137.36±3.99 | 272.70±13.85 | 289.35±9.95 |
| 16 | 124.77±4.34 | 134.43±3.98 | 266.57±8.81 | 280.89±7.73 |
| 17 | 122.46±3.74 | 134.39±3.38 | 261.08±6.31 | 280.97±4.28 |
| 18 | 121.05±3.02 | 133.16±2.85 | 256.08±4.42 | 277.83±4.16 |
| 19 | 119.78±2.47 | 131.98±2.79 | 252.44±2.96 | 276.09±3.38 |
| 20 | 118.98±3.12 | 130.84±2.88 | 252.58±3.33 | 275.44±6.96 |
| 21 | 118.17±1.43 | 130.31±2.21 | 251.10±1.94 | 276.82±5.68 |
| 22 | 117.41±1.66 | 130.09±1.35 | 250.62±3.28 | 275.24±2.85 |
| 23 | 117.10±1.50 | 129.51±0.70 | 250.37±3.73 | 275.98±5.10 |
| 24 | 117.31±1.70 | 129.64±0.95 | 250.08±2.63 | 274.88±3.05 |
| 25 | 117.29±2.03 | 130.12±1.66 | 252.79±7.29 | 276.21±6.28 |
| 26 | 116.57±1.74 | 128.37±1.14 | 251.36±4.30 | 273.79±3.57 |
| 27 | 116.53±1.72 | 129.39±1.85 | 251.95±4.66 | 273.68±4.33 |
| 28 | 117.33±1.55 | 129.53±2.07 | 254.94±6.50 | 274.85±3.48 |
| 29 | 118.01±2.27 | 129.83±0.74 | 253.98±5.62 | 274.99±2.18 |
| 30 | 116.78±1.41 | 129.02±2.83 | 254.26±3.44 | 274.49±4.31 |
| 31 | 116.21±1.78 | 130.98±1.48 | 258.09±5.12 | 279.86±6.69 |
| 32 | 117.82±2.26 | 132.01±3.26 | 252.32±0.48 | 282.04±6.78 |
| 33 | 119.97±1.03 | 131.50±1.11 | 254.69±7.24 | 275.95 |
| 34 | 118.34±0.78 | | 259.05±8.57 | |
| 35 | 117.76 | | 259.25 | |
| 36 | 117.51 | | 258.95 | |
| 37 | 118.48 | | 261.14 | |

excluded). We were unable to find height data for 1% of men and 2% of women and weight data for 3% of men and 8% of women.

As all data are publicly available, no ethical permission or informed consent was deemed necessary for this study.

## Statistical analyses

Descriptive statistics are presented as means and standard deviations of height, weight, PPA in the all-time top lists and annual records of all athletes in different ages in all events for men and women. To compare height, weight and PPA in different stroke groups and events, we also performed a one-way ANOVA and independent t-test to compare males and females, and former and recent swimmers (separately for every stroke or event). Cohen's d effect size (ES/η2) and confidence intervals (CI) were reported. Using peak performance trends (Table 1), an event- and sex-specific analysis was performed by analysing the collected data with unbalanced dynamic panel data models incorporating fixed effects. The parameters were estimated using the plm package (linear models for panel data) in R (version 3.6.2). The assumption of no autocorrelation was tested using the Arellano-Bond test, and the validity of the instruments was assessed with the Sargan test. In all events, the collected data were divided into training and test data sets. The training sample consisted of a randomly drawn set of 18 observations that were used to estimate the models. These estimated models were then used

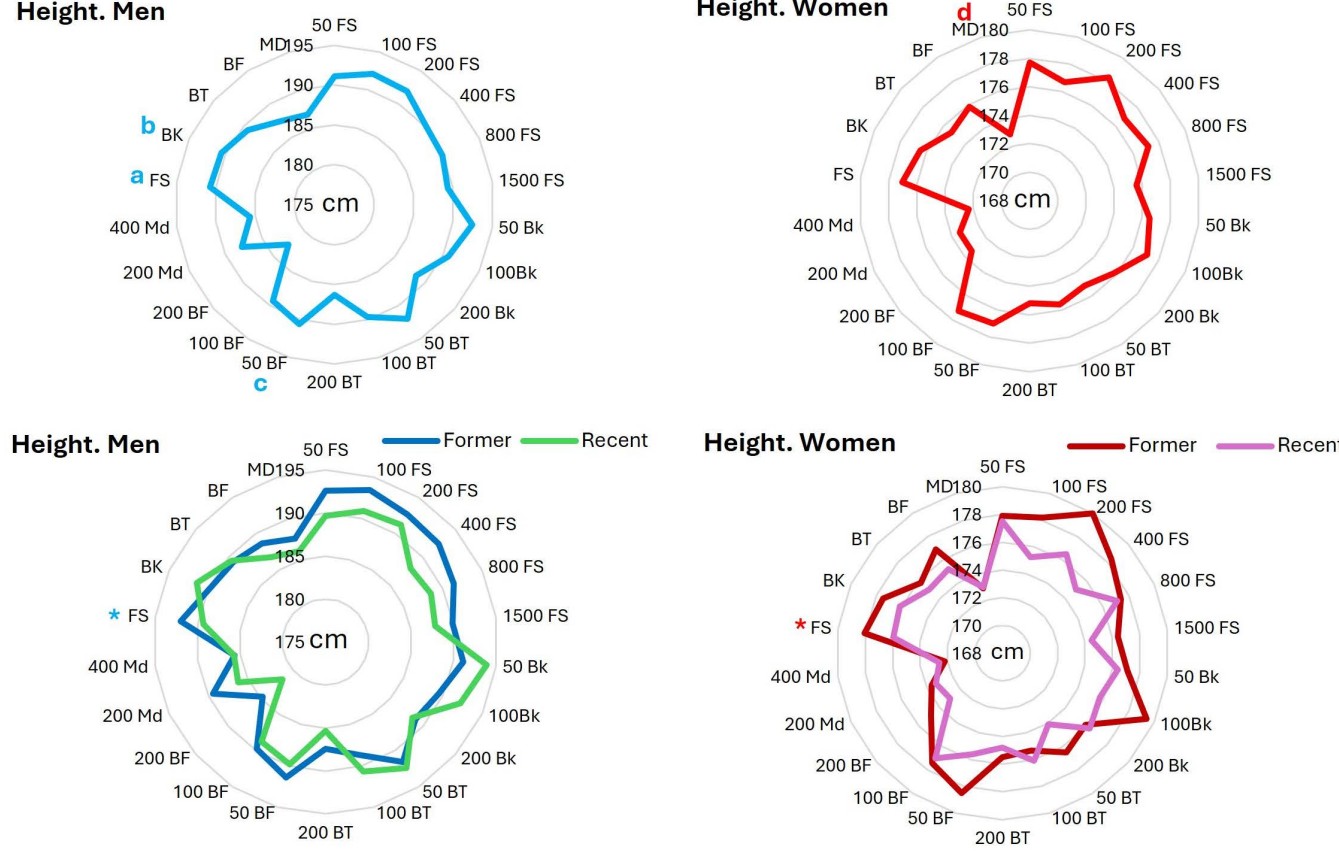

**Fig 1. Height of All-time men and women top lists in 5 styles and 17 events. a = significantly taller than BF and MD; b = significantly taller than MD; c = significantly taller than 200m BF; d = significantly shorter than FS, BK and BF.** * = significantly taller than their recent counterparts.

to calculate and plot 95% confidence bands. In addition, we plotted the two remaining test samples to further evaluate the obtained models. Since the test samples were not included in the modelling and the data was limited, based on the results of testing 90:10 and 80:20 in some disciplines as a pilot, we decided to prioritise 90:10 for the final presentation.

## Results

### Profile analysis of height and weight

1. **Height**: The analysis of the height of the male top list swimmers ($F_{4.332} = 5.17$, P = 0.001, ES = 0.104) showed that height of FS swimmers was significantly larger than that of BF (P = 0.015; CI = 0.0040 to 0.0566) and MD swimmers (P = 0.004; CI = 0.0090 to 0.0698) and height of BK swimmers was significantly larger than that of MD swimmers (P = 0.014; CI = 0.0053 to 0.0734). The data for the intergroup events showed that only male 50m BF swimmers' height was significantly (P = 0.008; CI = 0.010 to 0.138) larger than that of 200m BF (Fig 1). The analysis of the height of the top list female swimmers ($F_{4.328} = 4.60$, P = 0.001, ES = 0.098) showed that the height of MD swimmers was significantly lower than that of FS (P = 0.001; CI = −0.0680 to −0.0132), BK (P = 0.006; CI = −0.0687 to −0.0078) and BF (P = 0.042; CI = −0.0618 to −0.0007) swimmers (Fig 1).

The comparison between former and recent swimmers in different strokes using independent t test showed that former male and female FS swimmers were significantly (M: P = 0.005; CI = 0.0082 to 0.045, W: P = 0.047; CI = 0.0002 to 0.041)

taller than more recent FS counterparts (M: 192.05 vs. 189.2; W: 178.05 vs. 175.98-Fig 1). Although there was no significant difference between swimmers in different events, in 12 events for men and 13 events for women (out of 17 events), the former swimmers were taller (up to 4.3 cm for men and 3.7 cm for women) than their recent counterparts. However, the recent men's 50m (2.7 cm) and 100m BK (2.7 cm) and 100m BT (2 cm) swimmers were insignificantly taller than their former counterparts (Fig 1).

2. **Weight**: The analysis of the weight of the male top-list swimmers ($F_{4.332}$ = 3.4, P = 0.01, ES = 0.080) showed that the weight of MD swimmers was significantly lower than that of the FS (P = 0.004; CI = −9.46 to −1.26) and BT (P = 0.026; CI = −9.72 to −0.38). The data for the intergroup events showed that only the weight of the male 50m BT swimmers was significantly (P = 0.023; CI = 0.65 to 19.24) greater than that of 200m BT counterparts (Fig 2). The analysis of the weight of the female swimmers in the top lists ($F_{4.307}$ = 37.50, P = 0.003, ES = 0.096) showed that the weight of MD swimmers was significantly lower than that of all other swimmers; FS (P = 0.037; CI = −6.55 to −0.13), BK (P = 0.001; CI = −8.77 to −1.54), BT (P = 0.012; CI = −7.93 to −0.64) and BF (P = 0.039; CI = −7.29 to −0.11) swimmers (Fig 2).

The comparison between former and recent swimmers in different strokes using independent t test showed that former male FS swimmers were significantly (P = 0.003; CI = 1.53 to 7.58) heavier than their more recent counterparts (86.6 vs. 82.05). In females, former BK, BF and MD swimmers were significantly (BK: P = 0.008; CI = 1.11 to 7.45, BF: P = 0.047; CI = 0.078 to 6.87, MD: P = 0.003; CI = 2.10 to 9.11) heavier than more recent counterparts (BK: 67.7 vs. 63.4; BF: 85.9 vs. 82.4; MD: 63.38vs. 57.77). In 13 events for men and 16 events for women (out of 17 events) body mass of recent

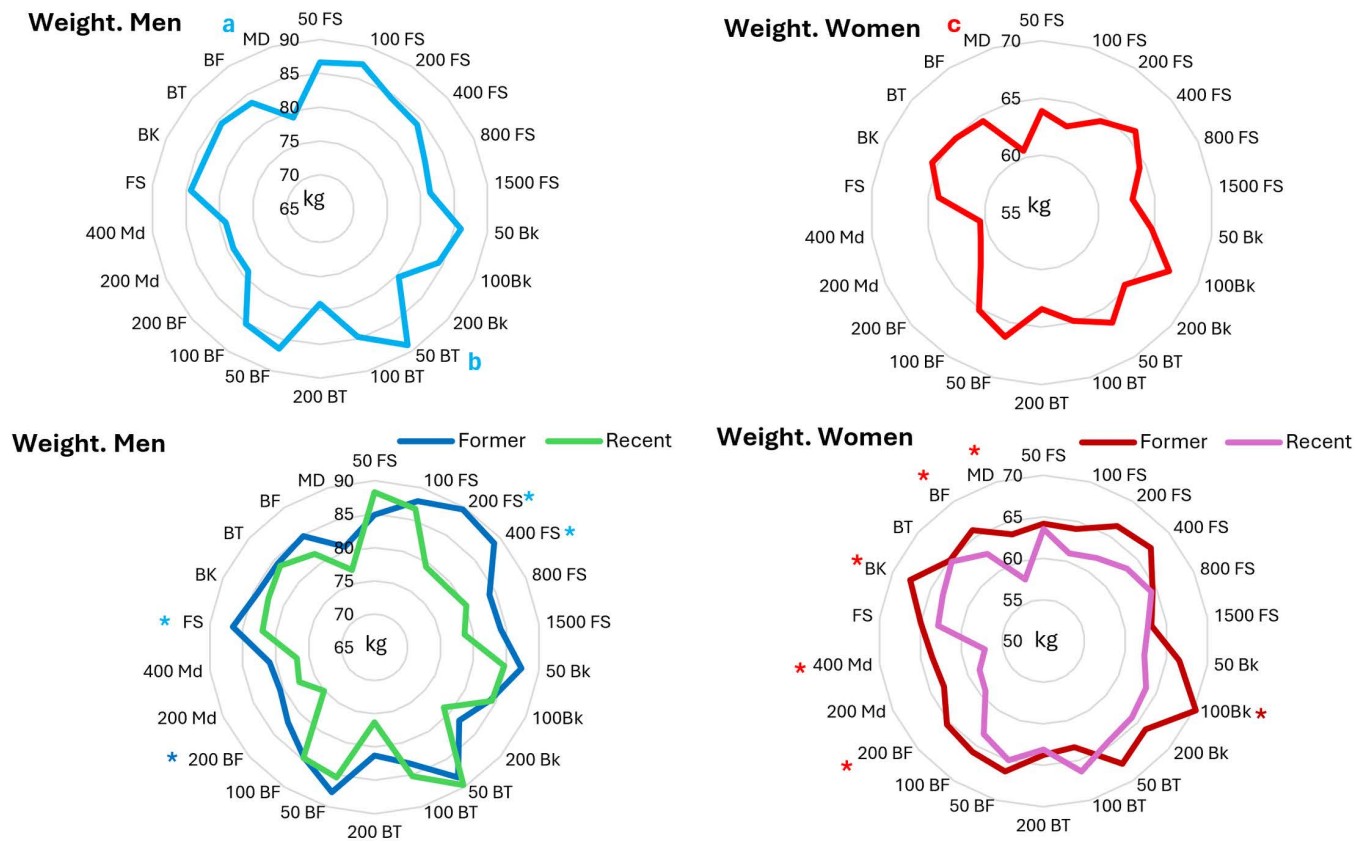

**Fig 2. Weight of All-time men and women top lists in 5 styles and 17 events. a = significantly lower than FS and BT; b = significantly higher than 200m BT; c = significantly lower than FS, BK, BT and BF.** * = significantly higher than their former counterparts.

swimmers was lower than that of former swimmers (up to 10 kg for men, and 6.4 kg for women; Men's 200m FS P = 0.008, 400m FS P = 0.01, 200m BF P = 0.005; and women's 50m BK P = 0.072, 100m BK P = 0.048, 200m BF P = 0.038, 200m MD P = 0.083, 400m MD P = 0.017) (Fig 2).

3. **Profile analysis of PPA:** The analysis of the PPA of the male top-list swimmers ($F_{4.335}$ = 4.76, P = 0.001, ES = 0.097) showed that the PPA of BT swimmers was significantly higher than that of FS (P = 0.001; CI= 0.57 to 3.05), BK (P = 0.002; CI= 0.50 to 3.36) and BF (P = 0.034; CI= 0.07 to 2.93) swimmers (Fig 3). Analysis of the PPA of the female top list swimmers ($F_{4.335}$ = 2.03, P = 0.089, ES = 0.054) showed that the PPA of the BF swimmers was significantly higher than that of the MD swimmers (P = 0.049; CI= 0.06 to 4.17) (Fig 3). Comparison of the data between events showed that the PPA of male 50m FS swimmers was significantly (P = 0.012; CI= 0.37 to 6.33) higher than that of 200m FS. Also, men's 50m BT swimmers PPA was significantly (P = 0.005; CI= 0.57 to 6.53) higher than 200m BT swimmers. In females, the PPA of 50m FS

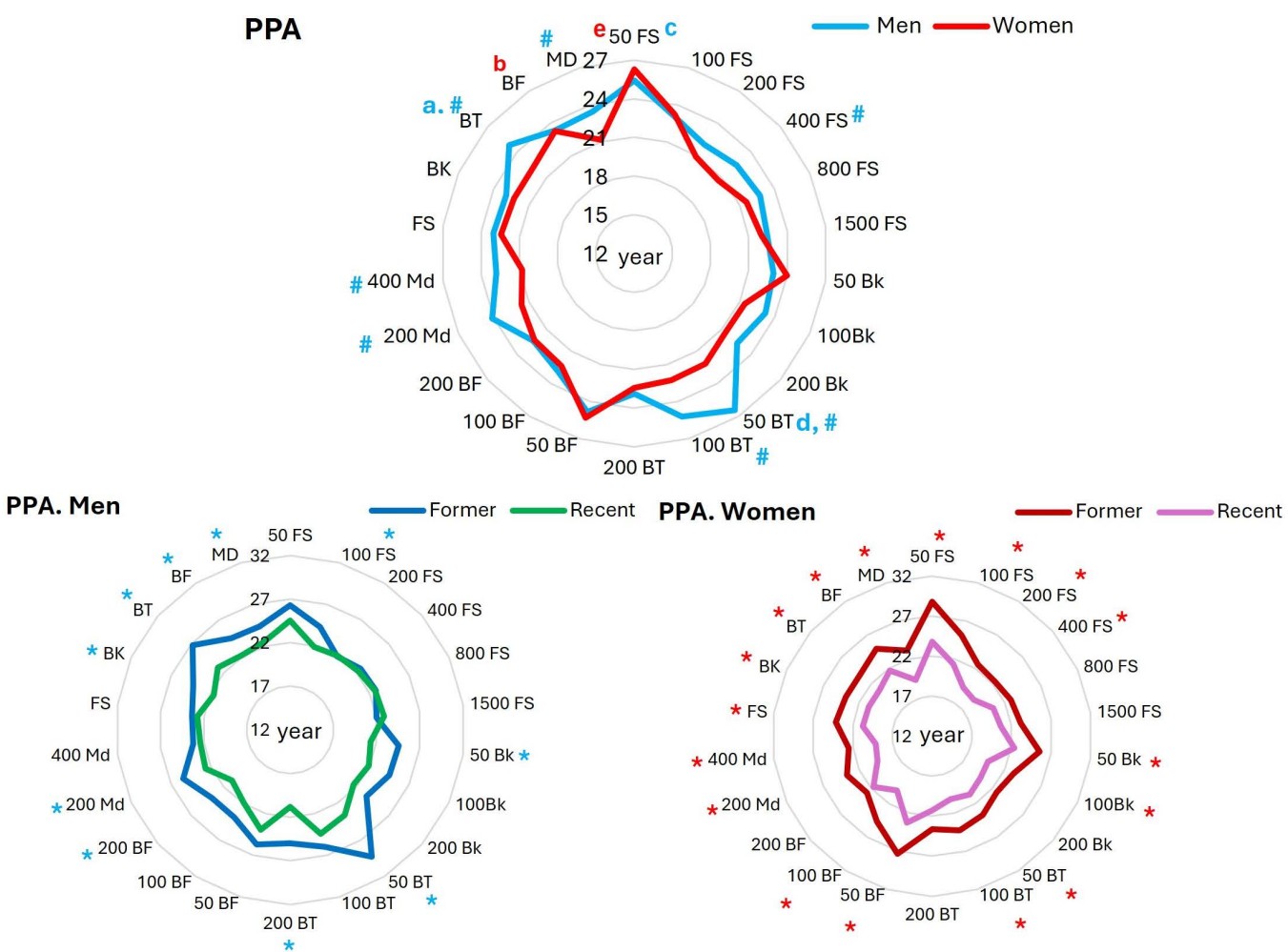

**Fig 3. Comparable PPA of All-time for men and women top lists in 5 styles and 17 events.** a = significantly higher than men's FS, BK and BF; b = significantly higher than women's MD; c = significantly higher than men's 200m FS; d = significantly higher than men's 200m BT; e = significantly higher than women's 200m, 400m 800m and 1500m FS. # = significantly higher than their women counterparts. * = significantly higher than their recent counterparts.

swimmers was significantly higher than 200m FS (P=0.001; CI=1.59 to 9.21), 400m FS (P=0.001; CI=1.84 to 9.46), 800m FS (P=0.003; CI=0.89 to 8.51) and 1500m FS (P=0.009; CI=0.54 to 8.16) swimmers (Fig 3).

On the other hand, the comparison of PPA between sexes in the different strokes and events showed that the PPA of males in the BT (P=0.001; CI:1.376 to 3.724) and MD (P=0.001; CI=15.645 to 21.089) strokes, and in 50m BT (P=0.001; CI=1.856 to 6.644), 100m BT (P=0.002; CI=1.166 to 4.734), 400m FS (P=0.018; CI=0.339 to 3.361), 200m MD (P=0.007; CI=0.738 to 4.262) and 400m MD (P=0.047; CI=0.024 to 3.976) events was significantly higher than females (Fig 3).

The comparison between former and recent swimmers in different strokes using independent t test showed that PPA of the former male BK, BT, BF and MD swimmers were significantly (BK: P=0.001; CI=1.4 to 3.86, BT: P=0.001; CI=2.39 to 5.27, BF: P=0.001; CI=0.92 to 3.67, MD: P=0.024; CI=0.024 to 3.35) higher than recent counterparts (BK: 24.23 vs. 21.6; BT: 26.76 vs. 22.93; BF: 24.5 vs. 22.2; MD: 24.35 vs. 22.55). In females, the PPA of the former swimmers in every stroke were significantly (FS: P=0.001; CI=2.1 to 4.7, BK: P=0.001; CI=1.35 to 4.91, BT: P=0.001; CI=1.73 to 4.53, BF: P=0.001; CI=1.45 to 5.01, MD: P=0.001; CI=2.09 to 5.5) higher than their recent counterparts (FS: 24.15 vs. 20.73; BK: 23.83 vs. 20.7; BT: 23.86 vs. 20.73; BF: 24.93 vs. 21.7; MD: 23.1 vs. 19.3). In 15 events for males and 17 events for females (out of 17 events) the PPA of recent swimmers was lower than the former swimmers (up to 5.7 years in men, and 5.7 years in women (Fig 3).

**4. Performance Progression Modelling:** Dynamic panel data for predicting future records with different coefficients (plots for the 95% confidence bands) in different events showed that performance over the preceding 1 year in all events and 2 years in men's 50m BT and women's 100m BK, 100m BT, 200m BT and 200m BF, and weight in 100m BK (P<0.05) were important (Fig 4).

## Discussion

The results of our study have shown that: 1) Performance over the preceding 1 year in all events and 2 years in men's 50m BT and women's 100m BK, 100m BT, 200m BT and 200m BF and body mass in 100m BK were important for modelling of future performance. 2) The FS and BK swimmers were significantly taller and heavier than the other swimmers, while the MD swimmers were significantly shorter and lighter; 3) Both male and female more recent swimmers were shorter, lighter and, in particular, younger than their former counterparts in most events.

Our prediction model for swimming showed no dependence of swimming events on the relevance height, which contradicts a previous athletics study for the long jump and triple jump where height was important [11], but consistent with multivariate analysis of performance in young swimmers [10]. It is possible that almost all top swimmers in a particular discipline have the required minimum height and that a higher stature does not bring any additional advantage. However, male and female 50m swimmers were taller than their 200m counterparts in almost all styles. On the other hand, the recent swimmers (in 12 of 17 events for men and 13 for women) were smaller (up to 4.3 cm for men and 3.7 cm for women) than their former counterparts. In contrast, the recent swimmers in the 50m and 100m BK and 100m BT for men were taller than their former counterparts. The considerable differences between the height of former and recent swimmers could be an additional reason why the height of all 20 top swimmers combined had no effect on the prediction model.

Our model showed that body mass is an important factor for predicting future performance only in women's 100m BK swimming. In line with this result, among all swimming strokes for both men and women, the 100m BK female swimmers were the only 100m swimmers who were heavier than their 50m counterparts in all men's and women's swimming strokes. As with height, the considerable variations in weight between former and recent swimmers could be an additional reason why the weight of all 20 top swimmers combined had no effect on the prediction model. Weight comparisons between former and recent swimmers showed that in most events recent swimmers are lighter than the former swimmers (up to 10 kg for men and 6.4 kg for women; statistically significant in 3 events for men and 3 events for women). It is possible that more advanced functional strength training and the avoidance of unnecessary

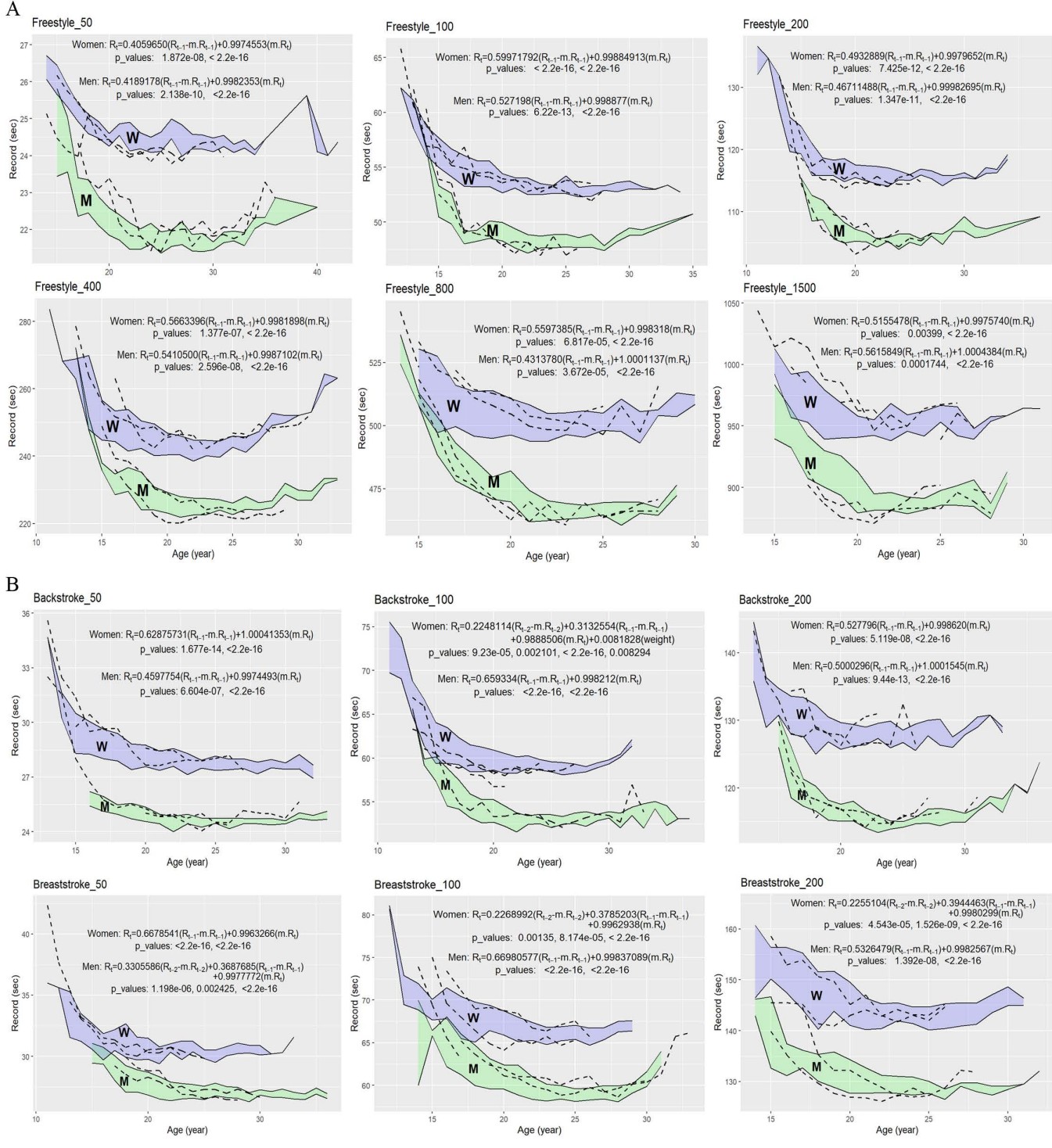

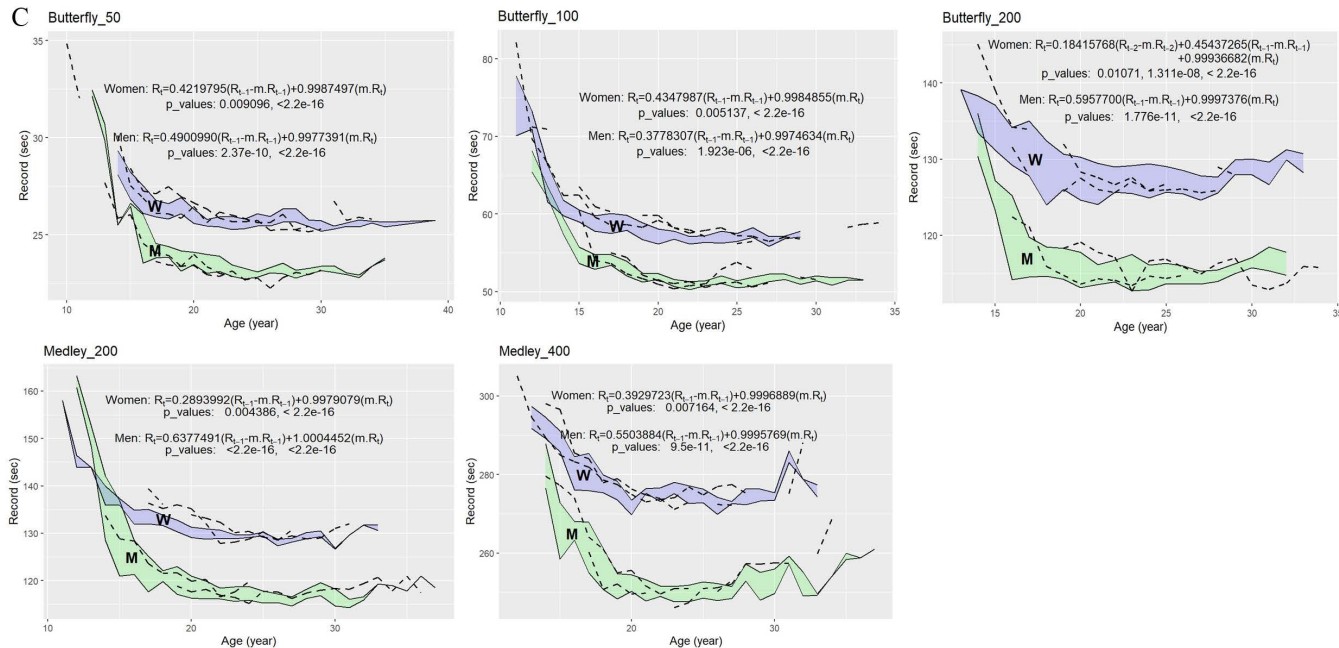

**Fig 4. The plots for the 95% confidence bands used to estimate the models using data from 18 swimmers.** Two remaining test patterns plotted with dashed lines for further evaluation of the obtained models. For example, future performances for a 16-year-old male swimmer in the 100 m freestyle with a height of 191 cm and a best performance of 53.21 s at age 15 are estimated to be 51.72, 49.47, 49.40, 49.40, 48.88, 48.49, and 48.16 s at ages 16, 17, 18, 19, 20, 21, and 22, respectively. Rt −1 is the best personal record one year ago, Rt −2 is the best personal record two years ago, Rt.m is the average (mean) of the records of the all-time top lists at a given age (Table 1). The factors for each discipline were set to a statistical significance of P < 0.05. The R software dynamic panel data P-values are indicated in each case.

muscle mass gain, as well as evolving refinement in optimal swimming technique, contribute to this observation. Interestingly, in the men's 50m FS, the most popular event in swimming, the recent swimmers were about 3.4 kg heavier but 3 cm shorter than their former counterparts (BMI; former: 22.9 kg/m² vs. recent: 24.5 kg/m²) indicating that muscle power is very important in this event. Contrary to the expectation of Charles and Bejan [20], it appears that modern fast swimmers will be smaller and lighter than in the past.

Gorzi et al. (2022) reported that the preceding two years of records were important for predicting next-year performance in about 63% of track and field disciplines for men and 71% for women [11], but in the current study, the preceding two years of best records were important for predicting next-year performance in only 6% of events for men and in 23% for women. It appears that in swimming, current/more recent performance (i.e., last year's best records) are more important and a long training history appeared to have no significant effect on current performance [10]. Wider confidence bands in the prediction of some events (200m vs. 50/100m) could be explained by the dependence on additional factors, such as recent training of the aerobic and anaerobic lactic systems or an appropriate pacing and distribution of effort in longer compared to shorter distances. [1]. Consequently, it is also more difficult to predict progression in longer swimming events, which may have implications for talent identification.

The results of our study showed that the PPA of top male BT swimmers was higher than that of FS, BK and BF swimmers, and the PPA of 50m FS swimmers was higher than that of 200m, 400m and 1500m FS swimmers, and the PPA of 50m BT swimmers was higher than that of 200m BT swimmers. It is plausible that the BT swimming style is more complicated and takes more years to master and improve timing and coordination than other styles. In women, the PPA of BF swimmers was higher than that of MD swimmers, and the PPA of 50m FS swimmers was higher than

that of 200m, 400m, 800m and 1500m FS swimmers. In agreement with the study of 2012 Olympic swimmers by Allens et al. [31], in the current study the PPA of men was higher than women in the BT and MD swimming, and in the 50m BT, 100m BT, 400m FS, 200m MD and 400m MD events. Swimmers with a lower PPA may have a limited window of opportunity to reach their best performance. The earlier onset of puberty in females compared to males could explain the earlier PPA in females [7,16,32,33]. Significant differences in PPA, particularly within the same competitive styles (e.g., freestyle), suggest that changing event distance with age may be an appropriate strategy [8]. However, it appears that in contrast to athletics, switching from higher distances to lower distances may be appropriate in swimming.

Peak performance was achieved at an older age for 50m swimmers than for 100m and 200m swimmers in almost all styles. This infers that coaches should have realistic expectations of the time required to reach peak performance in these events. It seems that the short distance events require more muscle power than the long distance events, and a higher number of years of training is needed to reach the required muscle mass for maximal strength [34]. The higher body mass of 50m and 100m swimmers in all FS, BK, BT and BF strokes compared to their 200m counterparts, and the higher BMI of 50m swimmers compared to the 100m and 200m swimmers in almost all swimming strokes for both men and women, support this claim. Swimmers who swim shorter distances may need to go through many training years after being fully mature to generate sufficient power to perfect their technique at swimming speeds close to competition speed [31,35].

Former and recent swimmers' comparisons showed that more recent male and female swimmers are younger than their former counterparts in PPA in almost all strokes (M: 4 out of 5, W: 5 out of 5) and all events (M 15 out of 17, W: 17 out of 17). Early specialization, more advanced swimming training and conditioning methods and improved recovery strategies and swimming equipment [11,36,37], along with secular trends in pubertal timing [23–25], might be possible reasons for this early PPA. The trend towards earlier PPA in elite swimmers could pose a risk of dropout and early retirement [38–40], which should be investigated further.

The results of the male and female top list swimmers showed that 93%–100% of the swimmers participated in more than one swimming stroke (e.g., backstroke and breaststroke). This result indicates that more varied swimming training can promote holistic development, which has a positive effect on performance in various swimming competitions. As the all-round development and training history of swimmers (training experience) is an important determinant of future performance and the window of peak performance [41], the limitation of not having access to this data is a limitation in our study. Another limitation is that we did not have access to swimmer's month of birth (relative age effect). Swimmers born at the beginning of the year might have advantages over those born at the end of the year [42,43]. The biological age of the swimmers is another consideration that were not included in this study [44]. Another limitation is that the prediction model was created by combining the results of more recent and former swimmers, with considerable differences between them. A comprehensive study on the differences in competition over shorter and longer distances in different sports and within sports in men and women is needed. Although the effects of genetic and environmental variables on swimming performance [45] and the nonlinear processes of growth and maturation [46] along with individual differences complicates the prediction process, it will provide a guideline for setting point targets for younger athletes.

## Practical applications

Although performance is influenced by many factors, such as the quality of training, nutrition, genetics, environment, and social and psychological factors [26], our event/sex-specific analysis for predicting records in swimming has shown that the last year's best record in all events and last 2 years' best records in men's 50m BT and women's 100m BK, 100m BT, 200m BT and 200m BF, and body mass in 100m BK, are important for predicting performance. Our model helps coaches with long-term planning by pointing to realistic targets for younger athletes. In addition, our study has shown that switching from longer to shorter distances in swimming can be useful to maintain a longer athletic career.

## Supporting information

**S1 Table. Models introduced by dynamic panel data using top list results.**
(DOCX)

**S2 Table. All data for top swimmers in different strokes and disciplines (height, weight, and records at different ages).**
(XLSX)

## Author contributions

**Conceptualization:** Amir Nazari Mehrabi, Hamoon Imani, Mina Khantan, Tommy R. Lundberg, Ali Gorzi.

**Data curation:** Amir Nazari Mehrabi, Hamoon Imani, Omid Khademnoe.

**Formal analysis:** Amir Nazari Mehrabi, Omid Khademnoe, Mina Khantan, Tommy R. Lundberg.

**Investigation:** Amir Nazari Mehrabi, Hamoon Imani, Mina Khantan, Tommy R. Lundberg, Ali Gorzi.

**Methodology:** Omid Khademnoe, Tommy R. Lundberg.

**Project administration:** Ali Gorzi.

**Resources:** Mina Khantan.

**Software:** Omid Khademnoe.

**Supervision:** Ali Gorzi.

**Validation:** Omid Khademnoe.

**Writing – original draft:** Mina Khantan, Tommy R. Lundberg, Ali Gorzi.

**Writing – review & editing:** Tommy R. Lundberg, Ali Gorzi.

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
