## [Decision Letter · Decision Letter 0]

3 Apr 2025

PONE-D-25-06170Modelling of performance prediction by analysis of elite swimmers’ anthropometry, peak performance age and age-related performance progressionPLOS ONE

Dear Dr. Gorzi,

Thank you for submitting your manuscript to PLOS ONE. After careful consideration, we feel that it has merit but does not fully meet PLOS ONE’s publication criteria as it currently stands. Therefore, we invite you to submit a revised version of the manuscript that addresses the points raised during the review process.

Please note that we have only been able to secure a single reviewer to assess your manuscript. We are issuing a decision on your manuscript at this point to prevent further delays in the evaluation of your manuscript. Please be aware that the editor who handles your revised manuscript might find it necessary to invite additional reviewers to assess this work once the revised manuscript is submitted. However, we will aim to proceed on the basis of this single review if possible.  Please pay particular attention to the expert reviewer's comments regarding greater depth of background literature required in the introduction, as well as greater methodological clarity.

We look forward to receiving your revised manuscript.

Kind regards,

Avanti Dey, PhD

Staff Editor

PLOS ONE

Journal Requirements:

Additional Editor Comments (if provided):

Reviewers' comments:

Reviewer's Responses to Questions

**Comments to the Author**

1. Is the manuscript technically sound, and do the data support the conclusions?

Reviewer #1: Yes

2. Has the statistical analysis been performed appropriately and rigorously? 

Reviewer #1: Yes

3. Have the authors made all data underlying the findings in their manuscript fully available?

Reviewer #1: No

4. Is the manuscript presented in an intelligible fashion and written in standard English?

Reviewer #1: Yes

5. Review Comments to the Author

Reviewer #1: A review of the manuscript titled:

Modelling of performance prediction by analysis of elite swimmers’

anthropometry, peak performance age and age-related performance progression

The manuscript presents an interesting study on the developmental progression of sports performance of world-class swimmers. The study is enriched by a comparison of basic somatic indicators of two generations of swimmers, which should be better explained.

In this manuscript, interesting conjectures are briefly discussed, without an attempt to deepen the literature, which makes these considerations shallow, burdened with subjectivity, although often accurate.

Precise, specific suggestions for organizing the content and necessary changes are below.

Specific comments:

Abstract

What did the authors have in mind when writing the sentence in lines 17-19? Please change, explain.

Introduction

Lane 43 There are more current considerations and studies on the development of young swimming talents in the literature, and this is an important issue. Therefore, omitting important references, with one Spanish one, is a simplification.

Lines 46-50 Reference to publications and specific tests - water and general - land in the prediction of swimming efficiency and development would be justified, the reviewer notes the omission of relevant literature considerations that exist.

Lines 51-53 Similarly, only one citation was used for authors from 2022. For example, a year earlier, one publication precisely on the topic of mediation between maturation, body physicality, and performance in female swimmers was published in the same journal, in others as well, later this year as well, which concerns swimming, not perception through other sports as in the manuscript.

Line 72, The insert regarding the aging aspect of swimming, is unnecessary without further involvement in this study.

Lines 76-79 the authors should have introduced at least some information regarding somatic changes of the new generation vs. the previous one instead of mentioning aging of swimmers, because aging itself is not the subject of considerations in this work.

2. Materials and Methods and Results

Lines 95-96 PPA, height and weight analysis was performed on the same subjects as the swimming results, or was some other historical analysis of a similar group used? Please explain.

Lines 128 - 222 This data should be inserted in an orderly manner into tables, and the progression of results as curves on charts. However, the attached charts should be of higher quality, because they are simply illegible.

4. Discussion

Lines 285 - 287 please explain what is important?

In lines 288-297 the focus should be on the results rather than considering the effectiveness of the prediction model used.

Lines 309 – 310 “The 100m BK female swimmers were the only 100m swimmers who were heavier than their 50m counterparts in all men’s and women’s swimming strokes.” - Please clarify.

Lines 322 – 323 References to biological age, puberty in girl swimmers and impact on athletic performance should be implemented.

Line 324 “free style” > freestyle

Similarly in line 337, without references to contemporary literature related to muscle mass and body composition of young swimmers.

Only somewhere around line 350-4 should considerations regarding the perfection of the prediction model be included to structure the discussion, reflecting on increasing anaerobic capacity, experience, perhaps pace distribution, etc.?

Line 358 (…) in multiple disciplines or in different swimming stroke races

Line 359 “history of swimmers (training age)“ > training experience, years of experience

Lines 361 - 363, if there is a mention of an age difference resulting from differences in the day, week, month in a given year, there should be a mention and a more current swimming reference to the differences in biological age.

References

References should be supplemented according to the suggestions for the appropriate sections of the manuscript.

6. PLOS authors have the option to publish the peer review history of their article (what does this mean? ). If published, this will include your full peer review and any attached files.

**Do you want your identity to be public for this peer review?** For information about this choice, including consent withdrawal, please see our Privacy Policy .

Reviewer #1: **Yes: ** Marek Strzała

---

## [Author Response · Author response to Decision Letter 1]

21 May 2025

PONE-D-25-06170

Modelling of performance prediction by analysis of elite swimmers’ anthropometry, peak performance age and age-related performance progression

Dear staff editor,

The authors acknowledge the constructive and positive feedback provided by the reviewer. Please see below our point-by-point response to the concerns, along with the revised manuscript with changes highlighted. We hope that the manuscript now qualifies for acceptance.

Sincerely,

The authors.

Reviewers' comments:

Reviewer's Responses to Questions

Comments to the Author

1. Is the manuscript technically sound, and do the data support the conclusions?

Reviewer #1: Yes

2. Has the statistical analysis been performed appropriately and rigorously?

Reviewer #1: Yes

3. Have the authors made all data underlying the findings in their manuscript fully available?

Reviewer #1: No We have added all data as supplementary 2.

4. Is the manuscript presented in an intelligible fashion and written in standard English?

Reviewer #1: Yes

5. Review Comments to the Author

Reviewer #1: A review of the manuscript titled:

Modelling of performance prediction by analysis of elite swimmers’

anthropometry, peak performance age and age-related performance progression

The manuscript presents an interesting study on the developmental progression of sports performance of world-class swimmers. The study is enriched by a comparison of basic somatic indicators of two generations of swimmers, which should be better explained.

In this manuscript, interesting conjectures are briefly discussed, without an attempt to deepen the literature, which makes these considerations shallow, burdened with subjectivity, although often accurate.

Precise, specific suggestions for organizing the content and necessary changes are below.

Specific comments:

Abstract

What did the authors have in mind when writing the sentence in lines 17-19? Please change, explain.

Thank you for this comment. We agree that it was unclear. Revised; Please see the revised version.

Introduction

Lane 43 There are more current considerations and studies on the development of young swimming talents in the literature, and this is an important issue. Therefore, omitting important references, with one Spanish one, is a simplification.

Thank you for this comment. We have added further references.

Lines 46-50 Reference to publications and specific tests - water and general - land in the prediction of swimming efficiency and development would be justified, the reviewer notes the omission of relevant literature considerations that exist.

Revised. Please see the revised version.

Lines 51-53 Similarly, only one citation was used for authors from 2022. For example, a year earlier, one publication precisely on the topic of mediation between maturation, body physicality, and performance in female swimmers was published in the same journal, in others as well, later this year as well, which concerns swimming, not perception through other sports as in the manuscript.

Thank you for pointing this out. Revised.

Line 72, The insert regarding the aging aspect of swimming, is unnecessary without further involvement in this study.

An important aspect of the paper is “age and the age of peak performance”. We noted, in the results and discussion, that more recent men and women swimmers reach their peak performance earlier than former swimmers (fig 3). Thus, discussing the age and aging factor is relevant in the manuscript.

Lines 76-79 the authors should have introduced at least some information regarding somatic changes of the new generation vs. the previous one instead of mentioning aging of swimmers, because aging itself is not the subject of considerations in this work.

Thank you for this very relevant comment. We have made revisions in both the introduction and the discussion.

2. Materials and Methods and Results

Lines 95-96 PPA, height and weight analysis was performed on the same subjects as the swimming results, or was some other historical analysis of a similar group used? Please explain.

This has been clarified in the revised version: (these 20 …).

Lines 128 - 222 This data should be inserted in an orderly manner into tables, and the progression of results as curves on charts. However, the attached charts should be of higher quality, because they are simply illegible.

All data are presented in a table (Suppl. 1 for models, and table 1 for progression of results). The figure resolutions improves if you click on the link to open the actual figure files.

4. Discussion

Lines 285 - 287 please explain what is important?

Thank you for this comment. Revised.

In lines 288-297 the focus should be on the results rather than considering the effectiveness of the prediction model used.

We agree, thank you for this comment. See revised version.

Lines 309 – 310 “The 100m BK female swimmers were the only 100m swimmers who were heavier than their 50m counterparts in all men’s and women’s swimming strokes.” - Please clarify.

We agree that this was unclear. Revised.

Lines 322 – 323 References to biological age, puberty in girl swimmers and impact on athletic performance should be implemented.

Revised.

Line 324 “free style” > freestyle

Revised.

Similarly in line 337, without references to contemporary literature related to muscle mass and body composition of young swimmers.

Revised.

Only somewhere around line 350-4 should considerations regarding the perfection of the prediction model be included to structure the discussion, reflecting on increasing anaerobic capacity, experience, perhaps pace distribution, etc.?

Revised.

Line 358 (…) in multiple disciplines or in different swimming stroke races

Revised.

Line 359 “history of swimmers (training age)“ > training experience, years of experience

Revised.

Lines 361 - 363, if there is a mention of an age difference resulting from differences in the day, week, month in a given year, there should be a mention and a more current swimming reference to the differences in biological age.

Revised.

References

References should be supplemented according to the suggestions for the appropriate sections of the manuscript.

Revised.

---

## [Decision Letter · Decision Letter 1]

24 Jul 2025

PONE-D-25-06170R1

Modelling of performance prediction by analysis of elite swimmers’ anthropometry, peak performance age and age-related performance progression

PLOS ONE

Dear Dr. Gorzi,

Thank you for submitting your manuscript to PLOS ONE. After careful consideration, we feel that it has merit but does not fully meet PLOS ONE’s publication criteria as it currently stands. Therefore, we invite you to submit a revised version of the manuscript that addresses the points raised during the review process.

We look forward to receiving your revised manuscript.

Kind regards,

Mário Espada, PhD

Academic Editor

PLOS ONE

Reviewers' comments:

Reviewer's Responses to Questions

**Comments to the Author**

1. If the authors have adequately addressed your comments raised in a previous round of review and you feel that this manuscript is now acceptable for publication, you may indicate that here to bypass the “Comments to the Author” section, enter your conflict of interest statement in the “Confidential to Editor” section, and submit your "Accept" recommendation.

Reviewer #2: All comments have been addressed

Reviewer #3: (No Response)

2. Is the manuscript technically sound, and do the data support the conclusions?

Reviewer #2: Yes

Reviewer #3: Yes

3. Has the statistical analysis been performed appropriately and rigorously? 

Reviewer #2: Yes

Reviewer #3: I Don't Know

4. Have the authors made all data underlying the findings in their manuscript fully available?

Reviewer #2: Yes

Reviewer #3: Yes

5. Is the manuscript presented in an intelligible fashion and written in standard English?

Reviewer #2: Yes

Reviewer #3: Yes

6. Review Comments to the Author

Reviewer #2: Congratulations on conducting the study. It advances the knownlege in the area. I understand that it is ready for publication.

Reviewer #3: This is a sound paper and with some further attention, could be a welcome addition to peer-reviewed evidence on the topic.

ABSTRACT

Line 35: “Both male and female more recent swimmers were shorter, lighter and, in particular, younger than their former counterparts in most events.” Without preceding content outlining the nature of the comparison, this makes little sense as presented. I.e. you do not allude to a “more recent” vs “less recent” (I suppose) comparison being an aim/part of methods.

INTRODUCTION

Well written, nothing to add.

METHODS

Mostly minor issues for consideration.

Line 118-119: Can you explain in a little more detail how you determined whether a swimmer was deemed former or recent?

Line 122-123: double semi-colon and no closing bracket.

Line 124: grammar… missing a “were included”?

Line 131-132: A little more detail is needed when you refer to data being “excluded by boxplot”… be more specific if you can.

Line 145-148: Can you add a citation to support your division of training and test set according to the 90:10 split you have adopted? Only because 80:20 would be more conventional.

Stat analysis: can you clarify the model used with the panel data models (i.e. fixed effects, random effects, other?). Were any assumptions checked (e.g. autocorrelation)? Re. the 18 observations, can you add how these were identified…randomly I guess? What statistics were used to evaluate the model fit (e.g. RMSE, AIC/BIC etc.). With so many independent samples t-tests executed, the authors are encouraged to consider protection against type I errors.

Table 1 suggests some missing data (i.e. gaps), how were these dealt with (statistically) exactly?

RESULTS

Line 153 and others: I feel presentation of your confidence intervals needs attention. E.g. “CI=.0040 .0566” would be better presented as “0.0040 to 0.0566”… repeat throughout results were applicable.

Line 152: you report an effect size here but there has been no mention of their use in the methods – can you go back and provide the details? I.e. are these partial eta-type ES stats or Cohen’s d? details about what is deemed small-large etc need adding too.

Line 154-155: “…and approaching significance for BF swimmers (P=0.53).”… this p-value is not approaching significance…missing zero? Regardless, it is not significant hence my suggestion is to remove this point (and any others about near significance like lines 164-169, 190-195). The results are already rather long, with many differences to comprehend, so this would aid interpretation for the reader.

Line 195: Linked to the above I guess, “were insignificantly heavier” in essence means they were not (statistically) heavier!

Lines 227-232: Suggest removal of all this content. Avoids repetition (i.e. we already know female differences from previous few lines), the details are cumbersome and many link to above comment re. non-sig. findings.

Performance progression modelling: Can you provide an indication of the model fit for each presented equation? Perhaps add them to the figures themselves.

DISCUSSION

First paragraph is more akin to a results subsection. Suggest re-write whereby you synthesise the key findings (i.e. that certain factors were of significance or not) rather than essentially listing every finding as you have.

Line 323: When stating 13/17 & 16/17 being lighter, this is based upon averages not the statistical findings hence it is exaggerated. Suggest re-writing with the numerator based on sig diffs only.

Line 341: Replace “It seems” with “It is plausible”.

Lines 373-374: This statement contradicts your panel data which regularly cited rt-1 and rt-2 (less often) as significant factors. Perhaps I have missed something here, but regardless, the point you are making needs attention so readers are not left confused. Indeed, on lines 401-403 you state “last year’s records in all events and last 2 years’ records in men’s 50m BT and women’s 100m BK, 100m BT, 200m BT and 200m BF, and weight in 100m BK, are important for predicting performance”.

General: for all the work that has gone into the production of the panel analysis, you have not overly discussed these findings. I.e. the balance of attention is on your comparisons (recent/former, male/female, t-test comparing events etc) with less emphasis on these plots, what they demonstrate and how that might aid practitioners. Can you do so?

---

## [Author Response · Author response to Decision Letter 2]

6 Aug 2025

Dear staff editor,

The authors acknowledge the constructive and positive feedback provided by the reviewer. Please see attached our point-by-point response to the comments, along with the revised manuscript with changes highlighted (also revised supp 1). We hope that the manuscript now qualifies for acceptance.

Sincerely,

The authors.

---

## [Decision Letter · Decision Letter 2]

29 Aug 2025

Modelling of performance prediction by analysis of elite swimmers’ anthropometry, peak performance age and age-related performance progression

PONE-D-25-06170R2

Dear Dr. Ali Gorzi,

We’re pleased to inform you that your manuscript has been judged scientifically suitable for publication and will be formally accepted for publication once it meets all outstanding technical requirements.

Kind regards,

Mário Espada, PhD

Academic Editor

PLOS ONE

---

## [Editor Report · Acceptance letter]

PONE-D-25-06170R2

PLOS ONE

Dear Dr. Gorzi,

I'm pleased to inform you that your manuscript has been deemed suitable for publication in PLOS ONE. Congratulations! Your manuscript is now being handed over to our production team.

Kind regards,

on behalf of

Dr. Mário Espada

Academic Editor

PLOS ONE